# Cofactor Self-Sufficient Whole-Cell Biocatalysts for the Relay-Race Synthesis of Shikimic Acid

Xiaoshuang Wang [1,2,3,†], Fengli Wu [2,3,*,†], Dan Zhou [2,3,4], Guotian Song [2,3], Wujiu Chen [2,3], Cuiying Zhang [1] and Qinhong Wang [2,3,*]

[1]  College of Biotechnology, Tianjin University of Science & Technology, Tianjin 300457, China; wangxs@tib.cas.cn (X.W.); cyzhangcy@tust.edu.cn (C.Z.)
[2]  CAS Key Laboratory of Systems Microbial Biotechnology, Tianjin Institute of Industrial Biotechnology, Chinese Academy of Sciences, Tianjin 300308, China; yuangauss@163.com (D.Z.); song_gt@tib.cas.cn (G.S.); chen_wj@tib.cas.cn (W.C.)
[3]  National Center of Technology Innovation for Synthetic Biology, Tianjin 300308, China
[4]  College of Bioengineering, Chongqing University, Chongqing 400044, China
*   Correspondence: wu_fl@tib.cas.cn (F.W.); wang_qh@tib.cas.cn (Q.W.)
†   These authors contributed equally to this work.

**Abstract:** Shikimic acid (SA) is a key intermediate in the aromatic amino-acid biosynthetic pathway, as well as an important precursor for synthesizing many valuable antiviral drugs. The asymmetric reduction of 3-dehydroshikimic acid (DHS) to SA is catalyzed by shikimate dehydrogenase (AroE) using NADPH as the cofactor; however, the intracellular NADPH supply limits the biosynthetic capability of SA. Glucose dehydrogenase (GDH) is an efficient enzyme which is typically used for NAD(P)H regeneration in biocatalytic processes. In this study, a series of NADPH self-sufficient whole-cell biocatalysts were constructed, and the biocatalyst co-expressing *Bmgdh–aroE* showed the highest conversion rate for the reduction of DHS to SA. Then, the preparation of whole-cell biocatalysts by fed-batch fermentation without supplementing antibiotics was developed on the basis of the growth-coupled L-serine auxotroph. After optimizing the whole-cell biocatalytic conditions, a titer of 81.6 g/L SA was obtained from the supernatant of fermentative broth in 98.4% yield (mol/mol) from DHS with a productivity of 40.8 g/L/h, and cofactor NADP$^+$ or NADPH was not exogenously supplemented during the whole biocatalytic process. The efficient relay-race synthesis of SA from glucose by coupling microbial fermentation with a biocatalytic process was finally achieved. This work provides an effective strategy for the biosynthesis of fine chemicals that are difficult to obtain through de novo biosynthesis from renewable feedstocks, as well as for biocatalytic studies that strictly rely on NAD(P)H regeneration.

**Keywords:** *Escherichia coli*; glucose dehydrogenase; shikimate dehydrogenase; cofactor regeneration; whole-cell biocatalysis; synthetic biology

## 1. Introduction

Shikimic acid (SA, 3,4,5-trihydroxy-1-cyclohexene-1-carboxylic acid) is a valuable compound, including six-carbon cyclitol with three chiral carbons and a carboxylic-acid functional group; thus, it can be used as an important precursor for the synthesis of many bioactive compounds [1–3]. In recent years, SA has gained great interest as a starting material for industrial synthesis of the anti-influenza drug oseltamivir phosphate (Tamiflu®). SA widely exists in a variety of plants, especially in the fruit of Chinese star anise (*Illicium verum*). However, there are many disadvantages of the plant extraction method, such as limitations of raw materials, low yield, a costly plant-based extraction procedure, and strong destruction to the natural environment [1,3]. Therefore, the traditional plant extraction method cannot meet the increasing demand for SA.

The shikimate pathway is a ubiquitous pathway in plants, algae, fungi, and bacteria, and it is the common pathway for organisms to synthesize various aromatic compounds [4–6]. The shikimate pathway consists of seven enzymatic steps, and SA is the metabolite of the fourth step. In the past two decades, many metabolic engineering approaches for constructing SA-overproducing *Escherichia coli* strains have been investigated and reported [7–11]. The strategies can be summarized as decreasing expenditure and increasing income by enhancing the supply of the two key precursors (PEP and E4P) of the shikimate pathway, broadening the metabolic flux of the shikimate pathway, and blocking the conversion of SA to downstream metabolites [6,12–14]. However, the de novo biosynthesis of SA is usually limited by intracellular NADPH supply.

The asymmetric reduction of 3-dehydroshikimic acid (DHS) to SA is catalyzed by shikimate dehydrogenase (EC 1.1.1.25) using NADPH as the cofactor. However, shikimate dehydrogenase catalyzes a reversible reaction of SA oxidation to DHS and DHS reduction to SA [15,16]. When NADPH is sufficient in the reaction system, it is conducive to converting DHS into SA. In contrast, SA tends to be oxidized into DHS [17]. Therefore, the biosynthesis of SA is directly related to the intracellular NADPH availability. Cui et al. overexpressed the membrane-bound transhydrogenase gene *pntAB* and the ATP-dependent NAD kinase gene *nadK* in the SA-producing *E. coli* SA112 strain, and the production of SA increased by 75.9% and 83.5%, respectively, while the intracellular NADPH concentration was more than doubled [18]. Kogure et al. found that the ratio of intracellular $NADPH/NADP^+$ in *Corynebacterium glutamicum* SKM4, SKM5, SKM6, and SKM7 strains gradually decreased with the titer of SA enhancement, and there was still about 20 g/L DHS in SKM7 fermentative broth that could not be converted into SA, indicating that the SA biosynthesis was limited by intracellular NADPH supply [1].

Since shikimate dehydrogenase requires NADPH as the cofactor, the conversion of DHS to SA may be continuously achieved by coupling an NADPH regeneration reaction with the DHS reduction process. This artificial biocatalytic approach was investigated in vitro through a multi-enzyme cascade system. First, DHS was synthesized from quinate via 3-dehydroquinate via two successive enzyme reactions, quinate dehydrogenase and 3-dehydroquinate dehydratase, with a yield of 57–77%. Then, shikimate dehydrogenase was coupled with glucose dehydrogenase (GDH, EC 1.1.1.47), and the intermediate DHS was completely converted into SA within 1.25 h in the presence of excess glucose (Table S1) [16]. Subsequently, the enzyme immobilization technology was adopted to further improve the biocatalytic efficiency for conversion of quinate to SA [19]. After that, Ghosh et al. utilized *Bacillus megaterium* as a whole-cell biocatalyst, achieving microbial transformation of quinate to SA with a maximum conversion ratio of 89%, but the substrate quinate concentration was only 5 mM (Table S1) [20]. Abdel-Hady et al. also achieved the efficient bioconversion of DHS to SA in vitro by coupling a thermostable phosphite dehydrogenase mutant $RsPtxD_{HARRA}$ with shikimate dehydrogenase [21]. Although the biocatalytic efficiency of this enzymatic system was high enough in vitro, the cofactor $NADP^+$ was exogenously supplemented and the substrate concentration was very low (Table S1). Thus, this strategy cannot meet the requirements of SA production on a large scale.

Since the efficiency of the NADPH regeneration reaction is relatively high, efficient bioconversion of DHS to SA may also be achieved using the residual small amount of intracellular $NADP^+/NADPH$. In order to verify this hypothesis, we co-expressed *aroE* and *gdh* in *E. coli* to prepare an NADPH self-sufficient whole-cell biocatalyst and systematically optimized the conditions of the whole-cell biocatalytic process. Finally, the efficient relay-race biosynthesis of SA from renewable glucose by coupling microbial fermentation with biocatalytic process was achieved.

## 2. Materials and Methods

### 2.1. Strains and Plasmids

The strains and plasmids used in this study are listed in Table S2.

## 2.2. Plasmid Construction

To construct protein expression plasmids, the glucose dehydrogenase genes derived from different organisms (GenBank accession no. NC_017138.1, AL445065.1, FJ908710.1, EF626962.1, and NC_014551.1) were artificially synthesized after optimizing the codon usage according to the bias of *E. coli*, and then cloned into the pETDuet-1 or pRSFDuet-1 plasmid at *Bam*HI and *Pst*I sites. The shikimate dehydrogenase gene *aroE* gene from *E. coli* BL21 (DE3) was amplified using aroE-NdeI-F and aroE-XhoI-R, and cloned into the pETDuet-1 or pRSFDuet-1 plasmid at *Nde*I and *Xho*I sites. When the order of *Bmgdh* and *aroE* was exchanged, the two gene fragments were amplified using Bmgdh-NdeI-F/Bmgdh-XhoI-R and aroE-BamHI-F/aroE-pstI-R, and cloned into the pRSFDuet-1 plasmid at *Nde*I-*Xho*I and *Bam*HI-*Pst*I sites, respectively. The phosphoglycerate dehydrogenase gene *serA* from *E. coli* BL21 (DE3) was amplified using serA-HR-F/serA-HR-R, and cloned into the pRSFDuet-Bmgdh-aroE plasmid at the *Xho*I site. The corresponding plasmid diagrams are provided in Figure S1. To construct pTargetF-serA, the pTargetF fragment with a targeting $N_{20}$ sequence of *serA* was obtained by inverse PCR using serA-N20-F/pTargetF-R, followed by self-ligation. All primers used in this study are listed in Table S3.

## 2.3. Deletion of Chromosomal serA

The chromosomal *serA* of *E. coli* BL21(DE3) was knocked out via the CRISPR/Cas9 system. The upstream and downstream homology arms of *serA* were amplified from the BL21(DE3) genome and then fused together by fusion PCR. The deletion procedures were conducted as described by Jiang et al. [22].

## 2.4. Media and Growth Conditions

*E. coli* DH5α was used as the cloning host. *E. coli* BL21(DE3) was used for recombinant protein expression and whole-cell biocatalysis. Cultures were grown at 30 °C or 37 °C in liquid Luria–Bertani (LB) medium (10 g/L tryptone, 5 g/L yeast extract, 10 g/L NaCl) or on agar plates. Corresponding antibiotics (100 µg/mL ampicillin, 50 µg/mL kanamycin, 50 µg/mL spectinomycin) were added into the culture medium as appropriate.

For protein expression in shake flask, a single colony was inoculated into 5 mL of LB medium with appropriate antibiotics and cultured overnight at 37 °C. The overnight seed culture was then inoculated into a 500 mL shake flask containing 100 mL of LB medium at a ratio of 1:100 and incubated at 37 °C, 250 rpm. When the culture reached an optical density of 0.6 at 600 nm, 0.2 mM isopropyl-β-D-thiogalactopyranoside (IPTG) was added to induce protein expression. After induction at 28 °C for additional 8 h, cells were harvested by centrifugation at 5000× *g* for 5 min at 4 °C.

For protein expression in a large scale, fed-batch fermentation was conducted in a 5 L bioreactor (BIOTECH-5BG, Bxbio, Shanghai, China). A single colony was inoculated into a falcon tube containing 5 mL of LB medium and cultured overnight at 37 °C. The overnight seed was then inoculated into a 1 L shake flask containing 200 mL of LB medium at a ratio of 1:100 and incubated at 37 °C, 250 rpm for 10–12 h. After this, the seed culture was transferred into 2.3 L of fermentation medium at a 1:12.5 (*v*/*v*) inoculum-to-medium ratio at an initial temperature of 37 °C. The fermentation medium was supplemented with appropriate antibiotics when required. The agitation, air supplementation, and feed rate were changed to maintain the dissolved oxygen (DO) concentration above 30% saturation. The pH was controlled at 7.0 using 25% (*w*/*v*) $NH_3·H_2O$. The DO-stat feeding strategy was employed to supply feeding medium to the fermenter. When the culture reached an optical density of 30–40 at 600 nm, the fermentation temperature was cooled to 30 °C, and then 0.5 mM IPTG was added to induce protein expression. Samples were collected every 2 h to determine cell density ($OD_{600}$), residual glycerol concentration, and plasmid stability. After induction at 30 °C for additional 8 h, cells were harvested by centrifugation at 6000 rpm for 15 min at 4 °C (Avanti JXN-26, Beckman, Brea, CA, USA). The fermentation medium contained 10 g/L glycerol, 2 g/L glucose, 10 g/L yeast extract, 16 g/L tryptone, 4 g/L $K_2HPO_4·3H_2O$, 2.24 g/L $NaH_2PO_4·2H_2O$, 3 g/L NaCl, 2.5 g/L $(NH_4)_2SO_4$, 2.1 g/L citric

acid monohydrate, 0.49 g/L MgSO$_4$·7H$_2$O, and 0.3 g/L FeSO$_4$·7H$_2$O. The feeding medium contained 600 g/L glycerol, 57.5 g/L yeast extract, and 92.5 g/L tryptone.

### 2.5. Enzyme Assays of GDH and AroE

About $5 \times 10^8$ cells were harvested by centrifugation at $8000 \times g$ for 2 min at 4 °C in a 2 mL centrifuge tube. Cell pellets were washed once with 0.5 mL of protein extraction buffer (100 mM Tris-HCl, pH 7.0, 20 mM KCl, 20 mM MgCl$_2$, 0.1 mM EDTA, and 2 mM DTT), and resuspended in 1 mL of the same buffer. The suspensions were sonicated for 5 min (200 W, pulse 3 s, interval 7 s) with an ultrasonic homogenizer (SCIENTZ-II D, Ningbo Scientz Biotechnology, China) in an ice-water bath. Cell debris was removed by centrifugation at $8000 \times g$ for 10 min at 4 °C, and the supernatants were used as crude extracts for enzyme assays. GDH activity was measured using a glucose dehydrogenase activity assay kit (Boxbio, Beijing, China) according to the instructions. AroE activity was measured as described previously [1,17] with some modifications. Briefly, the reactions were conducted in a 200 μL reaction solution consisting of 100 mM Tris-HCl buffer (pH 7.0), 1 mM NADPH, 1 mM DHS, and crude cell extract as appropriate. The enzymic activity was measured at 30 °C by monitoring the decrease in absorbance at 340 nm due to NADPH ($\varepsilon = 6.22 \times 10^3$ L/mol/cm) consumption. One unit of GDH activity (U) was defined as the amount of enzyme catalyzing the conversion of 1 μmol of NADPH per minute at 30 °C. Protein concentrations were measured using a Bradford protein assay kit (Beyotime Biotechnology, China) according to the instructions.

### 2.6. Whole-Cell Biocatalytic Conditions

For whole-cell biocatalysis by shake flask, a 10 mL scale reaction mixture consisting of 100 mM sodium phosphate buffer (pH 5.0–9.0), 0.1–20 OD$_{600}$ of whole-cell biocatalysts, 100 mM DHS (provided with crude fermentative broth), and 150 mM glucose was incubated at 30–40 °C and 200 rpm. For scale-up of whole-cell biocatalysis, the reaction mixture consisting of 1 L or 3 L of crude DHS fermentative broth (DHS titer > 80 g/L), 10–40 OD$_{600}$ of whole-cell biocatalysts, and 0.8–1.4 equivalents of glucose (molar ratio of glucose to DHS) was incubated in a 5 L bioreactor (BIOTECH-5BG, Bxbio, Shanghai, China). The whole-cell biocatalytic reaction was performed at 34 °C and 300 rpm for 3 h. The pH was maintained at 7.0 using 10 M NaOH.

### 2.7. Determination of DHS, SA, and Glycerol Concentrations in Fermentative Broth

Cells were removed from cultures by centrifugation, and culture supernatant was filtered using a membrane filter. The concentrations of DHS, SA, and glycerol were determined by HPLC (1200 series, Agilent Technologies, Palo Alto, CA, USA) equipped with a Rezex$^{TM}$ RFQ-Fast Acid H+ (8%) column (100 mm × 7.8 mm) (Phenomenex, Torrance, CA, USA). The analysis was performed at 55 °C with a mobile phase of 5 mM H$_2$SO$_4$ at a flow rate of 0.6 mL/min. DHS and SA were analyzed at wavelength of 210 nm, and glycerol was quantitated by refractive index detector (RID).

## 3. Results and Discussion

### 3.1. Screening GDHs to Construct NADPH Self-Sufficient Whole-Cell Biocatalysts

In our previous work, we constructed a DHS-overproducing *E. coli* strain WJ060 (Table S2) through metabolic engineering. The key genes involved in glucose transport, the glycolysis pathway, the pentose phosphate pathway, and the shikimate pathway were fine-tuned on the chromosome of *E. coli*. The rationally designed WJ060 strain grown in minimal medium produced ~90 g/L DHS through fed-batch fermentation with a total yield of 33.0% mol/mol glucose (Figure 1 and Figure S2). To construct the SA-overproducing strain, shikimate dehydrogenase gene *aroE* was overexpressed in WJ060 strain, combined with knocking out shikimate kinase genes *aroK* and *aroL*. However, only a small amount of SA was synthesized by this strain, and there was still a large amount of DHS in the fermentative broth. The intracellular NADPH supply limits the biosynthetic capability of

SA. GDH catalyzes the direct oxidation of glucose to gluconolactone using NAD(P)$^+$ as the co-substrate. Therefore, GDHs have been widely used for NAD(P)H regeneration in various biocatalytic systems [23–26]. If GDH is coupled with AroE in *E. coli* cells, DHS may be continuously converted into SA in the presence of excess glucose (Figure 1).

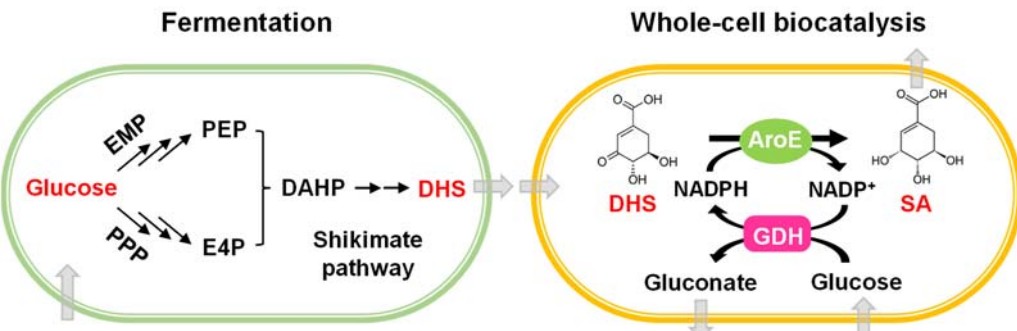

**Figure 1.** Simple diagram of the relay-race synthesis of SA from glucose by coupling microbial fermentation with whole-cell biocatalysis. EMP, Embden-Meyerhof-Parnas pathway; PPP, pentose phosphate pathway; PEP, phosphoenolpyruvate; E4P, D-erythrose 4-phosphate; DAHP, 3-deoxy-D-arabino-heptulosonate-7-phosphate; DHS, 3-dehydroshikimic acid; SA, Shikimic acid; AroE, shikimate dehydrogenase; GDH, glucose dehydrogenase.

In order to construct an efficient NADPH regeneration system, we screened several GDHs derived from different organisms, including *B. megaterium* [26], *Thermoplasma acidophilum* [27], *Lysinibacillus sphaericus* [28], *Bacillus subtilis* [29], and *Bacillus amyloliquefaciens* [30], and named them BmGDH, TaGDH, LsGDH, BsGDH, and BaGDH, respectively. Through SDS-PAGE analysis, most of these GDHs underwent soluble expression in *E. coli* BL21 (DE3), but the expression level of LsGDH was relatively low and TaGDH formed a large number of inclusion bodies (Figure S3). The enzyme activity assay showed that BmGDH activity was 58.9 U/mg, which was the highest activities among these five GDHs, while the activities of TaGDH and BaGDH were very low (Figure 2A). Although the expression level of LsGDH was lower than that of other GDHs, its enzyme activity was relatively high, indicating that the catalytic efficiency of LsGDH might be higher than that of the other GDHs. Only a small amount of BaGDH formed inclusion bodies, but its enzyme activity was still low (Figure 2A and Figure S3), suggesting that BaGDH possessed relatively poor catalytic efficiency.

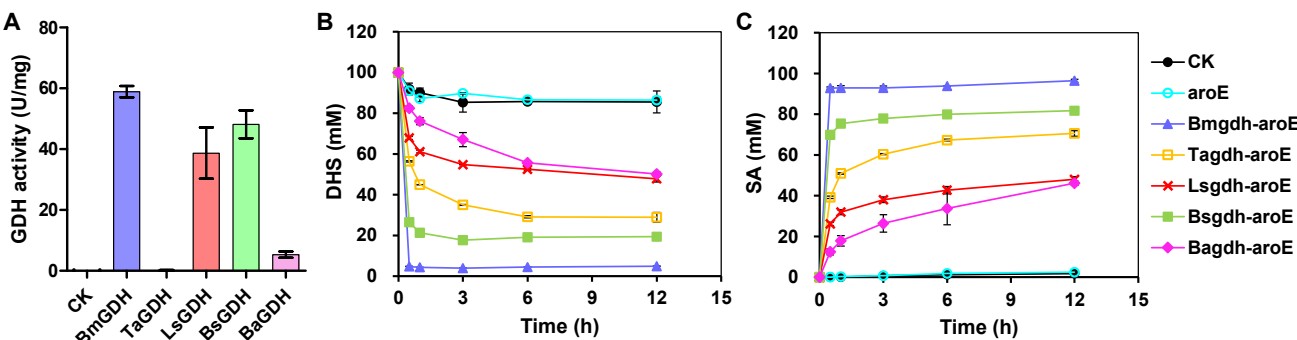

**Figure 2.** The effects of different GDHs on catalytic efficiencies of whole-cell biocatalysts. (**A**) GDH activities in the cell-free extracts of recombinant strains; Bioconversion of DHS (**B**) to SA (**C**) by whole-cell biocatalysts with different GDH–AroE combinations. The BL21(DE3) strain transformed with empty pETDuet-1 plasmid was used as the negative control (CK). Reaction conditions: 100 mM sodium phosphate buffer (pH 7.0), 100 mM DHS, 150 mM glucose, 20 OD$_{600}$ whole-cell biocatalyst, 10 mL of total volume, 37 °C. Data are presented as the mean ± standard deviation of three independent experiments.

These five *gdh* genes were co-expressed with *aroE* in *E. coli* BL21 (DE3) to obtain whole-cell biocatalysts. The whole-cell biocatalytic reactions were performed in shake flask using DHS fermentative broth of WJ060 strain as the substrate. The reaction rate of the BmGDH–AroE combination was the fastest, and about 92.8 mM SA could be synthesized after 0.5 h. The final DHS conversion rate of BmGDH–AroE was close to 100%, which was the highest yield among these five combinations. The yield of BsGDH–AroE combination was second, and the final conversion rate reached more than 80% (Figure 2B,C). Although TaGDH formed a large number of inclusion bodies and showed the lowest enzyme activities (Figure 2A and Figure S3), the TaGDH–AroE combination still achieved a conversion rate of more than 70% (Figure 2B,C). It might be that the co-expression of *Tagdh* and *aroE* led to some soluble expression of TaGDH, which led to TaGDH–AroE biocatalyst exhibiting activity. These results demonstrate that BmGDH–AroE is an attractive combination for the synthesis of SA. Therefore, the BmGDH–AroE combination was selected for subsequent whole-cell biocatalytic studies, and the BL21(DE3) strain with the pETDuet-Bmgdh-aroE expression plasmid was named PET. Through GDH screening, an efficient whole-cell biocatalytic system was constructed to achieve the relay-race biosynthesis of SA from renewable glucose (Figure 1).

In *E. coli*, the primary route for glucose uptake is the phosphoenolpyruvate/carbohydrate phosphotransferase system (PTS). Glucose is concomitantly phosphorylated during transport to produce intracellular glucose 6-phosphate, and then enters the glycolysis pathway and pentose phosphate pathway [31,32]. However, the direct substrate of GDH is glucose, not glucose 6-phosphate [23,33]. Therefore, co-expression of the non-PTS-dependent transporter gene may help to further improve the catalytic efficiencies of whole-cell biocatalysts. We co-expressed *Bmgdh–aroE* with *galP* (encoding galactose permease) derived from *E. coli* or *glf* (encoding glucose facilitator) derived from *Zymomonas mobilis* in BL21(DE3) to obtain whole-cell biocatalysts [32,34]. Unfortunately, the expression of *galP* or *glf* had no significant effect on the catalytic reaction rate and the DHS conversion rate (Figure S4), indicating that the process of glucose uptake was not the limiting factor for whole-cell biocatalysis.

### 3.2. Optimizing Catalytic Conditions for Efficient Whole-Cell Biocatalysis

In order to further improve the catalytic efficiency of the whole-cell biocatalyst, the reaction conditions were systematically optimized. The dose of the whole-cell biocatalyst loaded into the reaction is an important factor that influences catalytic efficiency and cost feasibility. When the PET biocatalyst load was less than 1 $OD_{600}$, the catalytic efficiency of the whole-cell biocatalyst was very low, which might be because the biocatalyst load was too low to meet the needs of the biocatalytic reaction. When the biocatalyst load reached 3 $OD_{600}$, the catalytic reaction rate was eight times higher than that at 1 $OD_{600}$. The SA titer for 3 $OD_{600}$ reached 63.4 mM at 0.5 h, and the productivity for each $OD_{600}$ biocatalyst was up to 42.3 mmol/L/h (7.4 g/L/h). When the biocatalyst load increased to 5–15 $OD_{600}$, SA production gradually increased with the increase in biocatalyst dose. There was no significant difference in yield between 15 $OD_{600}$ and 20 $OD_{600}$ (Figure 3A). These results demonstrate that the BL21(DE3) strain co-expressing *Bmgdh* and *aroE* is an efficient and attractive whole-cell biocatalyst with a suitable dose for the bioconversion of DHS to SA.

The effects of different pH on catalytic efficiency of whole-cell biocatalysts were also investigated between pH 5.0 and 9.0. With the increase in pH value, the DHS conversion rate firstly increased and then decreased, and the optimum pH for the whole-cell biocatalytic reaction was 7.0 (Figure 3B). It was reported that the optimum pH for the reduction of DHS to SA catalyzed by AroE derived from *C. glutamicum* was 7.0 [35]. In another report, the optimum pH for immobilized shikimate dehydrogenase (SKDH) was also 7.0, while that for GDH was 8.0–9.0. When SKDH was coupled with GDH, the optimum pH was also 7.0 [19], consistent with the whole-cell biocatalytic result obtained in this study.

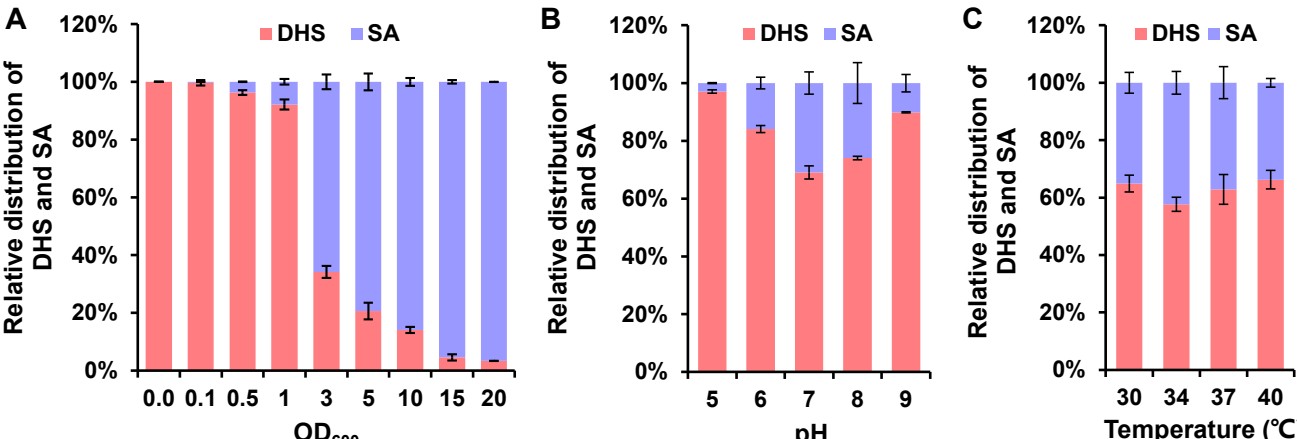

**Figure 3.** Effects of $OD_{600}$ (**A**), pH (**B**), and temperature (**C**) on bioconversion of DHS to SA by the PET whole-cell biocatalyst. The biocatalyst load for the whole-cell biocatalytic process was tested between 0.1 and 20 $OD_{600}$ at 37 °C, pH 7.0. The pH optimum of the whole-cell biocatalytic process was determined between pH 5.0 and 9.0 using 100 mM sodium phosphate buffer at 37 °C, with 2 $OD_{600}$ of PET whole-cell biocatalyst. The temperature optimum of the whole-cell biocatalytic process was determined at various temperatures (30–40 °C) at pH 7.0, with 2 $OD_{600}$ of PET whole-cell biocatalyst. All reactions were performed in 10 mL of 100 mM sodium phosphate buffer using 100 mM DHS and 150 mM glucose as substrates. Data are presented as the mean ± standard deviation of three independent experiments.

Glucose is oxidized by GDH to gluconolactone, which spontaneously hydrolyzes to gluconate, resulting in a decrease in pH value. When the initial pH was 5.0 or 6.0, the pH value of the catalytic system decreased gradually with the reaction proceeding, which was not conducive to the whole-cell biocatalytic process. When the initial pH was 8.0 or 9.0, the yield of SA decreased and the titer of gallic acid (GA) increased significantly (Figure 3B and Figure S5). The reaction catalyzed by AroE is a reversible reaction. It was reported that AroE tends to catalyze the reduction of DHS to SA at pH 7.0 and the oxidation of SA to DHS at pH 9.0 [35]. When assaying the enzyme activity of 3-dehydroquinate dehydratase/shikimate dehydrogenase (DQD/SDH), Huang et al. found that GA was mainly generated from DHS through nonenzymatic conversion in vitro at pH 9.0 [17]. However, in other studies, it was confirmed that shikimate dehydrogenase was involved in the biosynthesis of GA from DHS or SA [36,37]. The mechanism is likely that because SA can be oxidized to DHS and then to an intermediate 3,5-dehydroshikimate (3,5-DHS) catalyzed by shikimate dehydrogenase. The intermediate 3,5-DHS immediately and spontaneously converts to GA, as the enolization of 3,5-DHS to GA is energetically favorable [36]. The common conclusion from these studies is that DHS is easily converted to GA under alkaline conditions (such as pH 9.0). Therefore, when the whole-cell biocatalytic reaction is conducted at pH 7.0, it is not only beneficial to obtain the maximum reaction rate and SA production, but it can also inhibit the formation of byproduct GA.

The catalytic efficiency of the whole-cell biocatalyst was also influenced by temperature. In the range of 30–40 °C, the DHS conversion rate increased firstly and then decreased with the increase in reaction temperature, and 34 °C was the optimum temperature for the whole-cell biocatalytic reaction (Figure 3C).

### 3.3. Effect of the Expression Order of Bmgdh and aroE on Whole-Cell Biocatalytic Efficiency

The gene expression level is another important factor that affects the catalytic efficiency of whole-cell biocatalysts. Therefore, we investigated whether the order of *Bmgdh* and *aroE* in the pRSFDuet-1 plasmid affect their expression levels (Figure 4A). Through enzyme assays of BmGDH and AroE extracted from the cells co-expressing *Bmgdh–aroE* or *aroE–Bmgdh*, the BmGDH activity of the former was one-half that of the latter, while the AroE

activity of the former was eight times that of the latter (Figure 4B). To elucidate the reason for the difference in enzyme activity, we compared the protein expression levels of BmGDH and AroE. Since the molecular weights of BmGDH (28.07 kDa) and AroE (29.42 kDa) were almost equal, *Bmgdh* and *aroE* were solely cloned into the pRSFDuet-1 plasmid at the first or second expression site. SDS-PAGE analysis showed that the expression level of *aroE* at the second site was significantly higher than that at the first site, which was consistent with the result of the AroE enzyme assay (Figure 4B and Figure S6). This result suggests that the order of *Bmgdh* and *aroE* in the pRSFDuet-1 plasmid has a significant effect on the protein expression levels and enzyme activities. Gong et al. reported that the expression order of *LbCR*$_{M8}$ and *BmGDH* in the pETDuet-1 plasmid also had a significant effect on the enzyme activities. The activities of LbCR$_{M8}$ and BmGDH extracted from LbCR$_{M8}$-BmGDH cells were two and four times that of LbCR$_{M8}$ and BmGDH extracted from BmGDH-LbCR$_{M8}$ cells, respectively. After expression optimization, the whole-cell biocatalyst load was reduced about 27-fold [24].

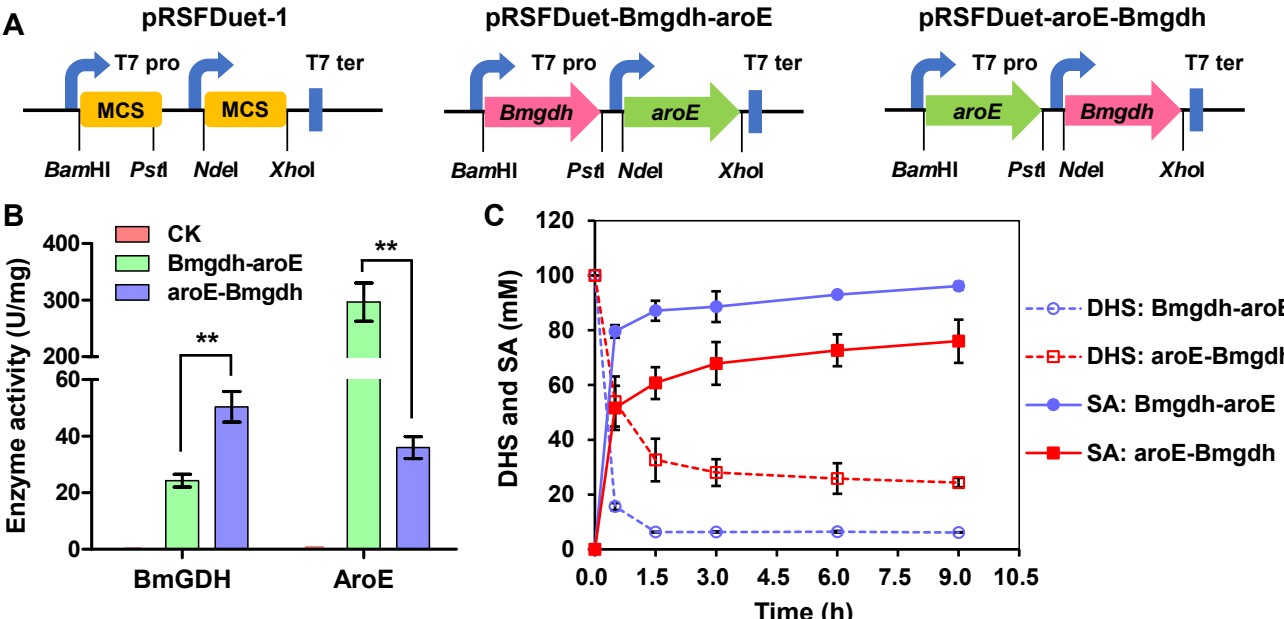

**Figure 4.** The effect of expression order for *Bmgdh* and *aroE* on the catalytic efficiency of whole-cell biocatalysts. (**A**) Simple diagrams of the plasmids for pRSFDuet-1, pRSFDuet-Bmgdh-aroE, and pRSFDuet-aroE-Bmgdh; (**B**) enzyme activities of BmGDH and AroE in the cell-free extracts of recombinant strains. The BL21(DE3) strain transformed with empty pRSFDuet-1 plasmid was used as the negative control (CK); (**C**) time courses of DHS and SA concentrations catalyzed by two types of whole-cell biocatalysts co-expressing *Bmgdh–aroE* or *aroE–Bmgdh*. The cells were harvested from shake flask cultures. Reaction conditions: 100 mM sodium phosphate buffer (pH 7.0), 100 mM DHS, 150 mM glucose, 5 OD$_{600}$ whole-cell biocatalyst, 10 mL of total volume, 34 °C. Data are presented as the mean ± standard deviation of three independent experiments. ** $p < 0.01$.

Although the expression order of *Bmgdh* and *aroE* on pRSFDuet-1 plasmid affected their expression levels, the whole-cell biocatalyst exhibited the best catalytic activity only when the activities of BmGDH and AroE were reasonably matched. When comparing the catalytic activities of the above two whole-cell biocatalysts, the reaction rate of BmGDH–AroE was significantly faster than that of AroE–BmGDH. At 0.5 h, the SA production for BmGDH–AroE reached 79.6 mM, while the production for AroE–BmGDH was only 51.6 mM. The conversion rate of the BmGDH–AroE was always higher than that of AroE–BmGDH throughout the whole catalytic process (Figure 4C). These results indicate that the reaction catalyzed by AroE may be the rate-limiting step for the whole-cell biocatalytic

process, and increasing the expression level of *aroE* is helpful to accelerate the reaction rate and improve the DHS conversion rate.

### 3.4. Optimizing Co-Expression System for Efficient Whole-Cell Biocatalysis

In order to prepare a large quantity of biocatalysts for whole-cell biocatalysis on a large scale, we examined the cell growth of three engineered strains harboring different *Bmgdh–aroE* expression plasmids by fed-batch fermentation. The PET strain harboring the pETDuet-Bmgdh-aroE plasmid grew fast, and the final $OD_{600}$ was up to 76.2 after fermentation for 14.5 h (Figure 5A). The *bla* gene on the pETDuet-1 plasmid encodes a secretory $\beta$-lactamase which catalyzes the hydrolysis of the ampicillin, leading to loss of its antibacterial activity [38]. Therefore, when adding IPTG to induce protein expression, additional 100 µg/mL ampicillin was supplemented to prevent plasmid curing from the cells. Unfortunately, the plasmid elimination rate was still very high, and only 4.1% of the cells retained ampicillin resistance (Figure 5D). The RSF strain harboring the pRSFDuet-Bmgdh-aroE plasmid was better than the PET strain, and about 83.3% of the cells maintained resistance to kanamycin at the end of fed-batch fermentation (Figure 5D). The cell grow rate of RSF strain was slower than that of the PET strain, and its $OD_{600}$ was as high as 72.1 after cultivation for 18.5 h (Figure 5B). These results demonstrate that replacing ampicillin with kanamycin was helpful to maintain intracellular plasmid stability during the fed-batch fermentation process.

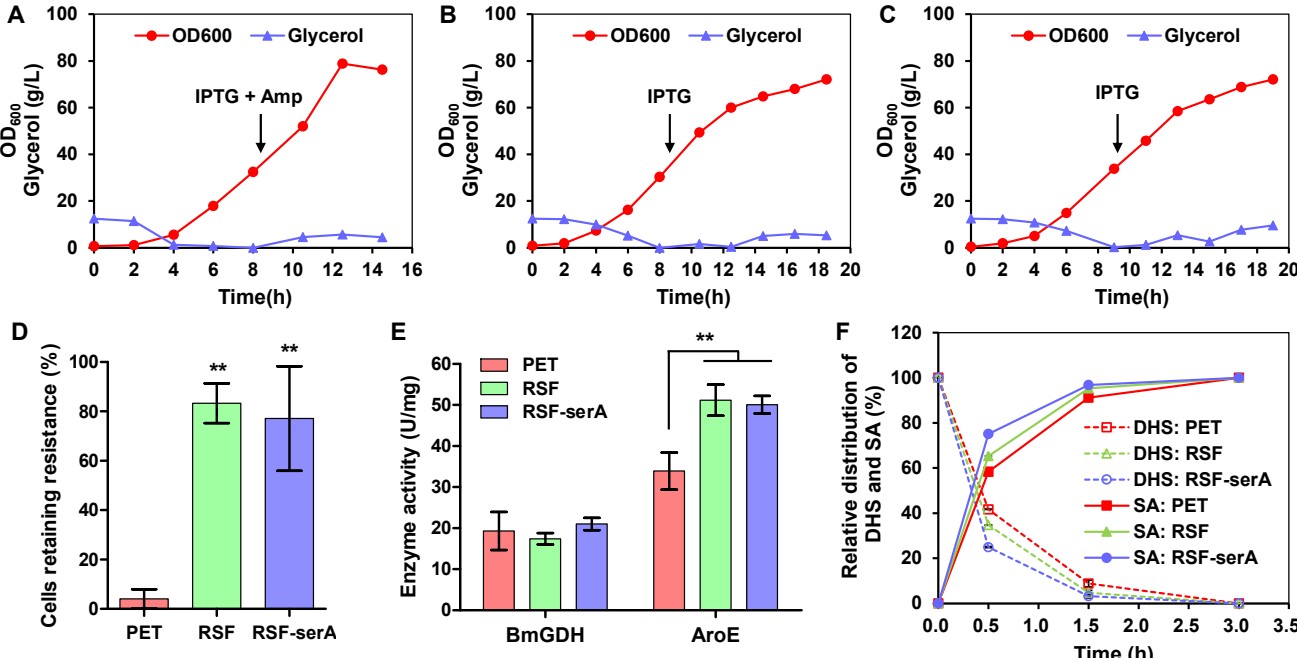

**Figure 5.** The effect of expression system on the catalytic efficiency of whole-cell biocatalysts. Time courses of cell growth (red circles) and glycerol concentration (blue triangles) for PET (**A**), RSF (**B**), and RSF-serA (**C**) strains by fed-batch fermentation. Fed-batch fermentation for each strain was conducted at least three times in a 5 L bioreactor, and the data shown in the figure are the results of one run; (**D**) the percentage of cells retaining antibiotic resistance. At least 50 colonies were examined in each experiment; (**E**) enzyme activities of BmGDH and AroE in the cell-free extracts of recombinant strains. Data in (**D**) and (**E**) are presented as the mean $\pm$ standard deviation of three independent experiments. ** $p < 0.01$. (**F**) Time courses of DHS and SA distributions catalyzed by three types of whole-cell biocatalysts. Reaction conditions: 80 g/L DHS, 1.4 equivalents of glucose, 20 $OD_{600}$ whole-cell biocatalyst, 1 L of total volume, 34 °C, pH 7.0. Data are presented as the mean $\pm$ standard deviation of two independent experiments.

The addition of antibiotics not only increases the cost of raw materials for preparing whole-cell biocatalysts and the cost of separation and purification for downstream products, but also causes environmental pollution due to the residual antibiotics in fermentative broth. The use of antibiotics can be avoided if auxotrophic strains are used as host cells. 3-Phosphoglycerate dehydrogenase (SerA) is the key enzyme in the L-serine biosynthetic pathway, and the *serA* deletion strains cannot synthesize L-serine, which is an essential amino acid for cell growth. According to previous reports, when *serA* is co-expressed with target genes by a plasmid using *serA* deletion strain as the host, the plasmid will self-replicate with the proliferation of the host cells without supplementing antibiotics [7,39,40]. The *serA* gene together with its own promoter was cloned into the pRSFDuet-Bmgdh-aroE plasmid at the *Xho*I site to form the pRSFDuet-Bmgdh-aroE-serA plasmid. This recombinant plasmid was then transformed into BL21(DE3) Δ*serA* strain to obtain the RSF-serA strain. The $OD_{600}$ of the RSF-serA strain was 72.1 after cultivation for 19 h, and its growth rate was almost the same as that of the RSF strain (Figure 5B,C). There was no kanamycin addition throughout the fed-batch fermentation process for the RSF-serA strain. Finally, about 77.1% of cells maintained resistance to kanamycin, which was almost comparable to that of the RSF strain (Figure 5D). If this strategy is applied to the study of whole-cell biocatalyst preparation, it can not only avoid the addition of antibiotics in fed-batch fermentation, but also achieve plasmid self-replication with the proliferation of the host cells.

To compare the catalytic efficiency of PET, RSF, and RSF-serA biocatalysts, the enzyme activity assay and scale-up of whole-cell biocatalysis were performed. There was no significant difference in BmGDH activities among three biocatalysts, but the AroE activities of RSF and RSF-serA were about 1.5 times that of PET (Figure 5E). However, compared with the result of Figure 4B, the activities of BmGDH and AroE for the RSF strain decreased significantly, which might be caused by the changes in culture conditions, such as fermentation mode, induction temperature, IPTG dose, plasmid stability, etc. Then, the catalytic efficiencies of the above three whole-cell biocatalysts were evaluated using DHS fermentative broth with high titer (>80 g/L) as the substrate in three bioreactors. At 0.5 h, the order of the reaction rate for three biocatalysts was RSF-serA >RSF > PET, and their DHS conversion rates reached 75.1%, 65.3%, and 58.3%, respectively. When the reaction proceeded to 3.0 h, the DHS conversion rates of the three biocatalysts were all close to 100% (Figure 5F). These results indicate that the whole-cell biocatalysts co-expressing *Bmgdh–aroE* were highly efficient for the bioconversion of DHS to SA. By optimizing the co-expression system, we not only developed a simplified fed-batch fermentation approach for preparing whole-cell biocatalysts, but also constructed an antibiotic-free protein expression system based on the growth-coupled L-serine auxotroph.

### 3.5. Optimizing Catalytic Conditions for Efficient Whole-Cell Biocatalysis on a Large Scale

In order to reduce the cost of production and separation for SA, we further optimized the load of biocatalyst and glucose for large-scale whole-cell biocatalysis. Under the conditions of shake flask biocatalysis, about 76 mM SA was synthesized by 5 $OD_{600}$ whole-cell biocatalyst within 0.5 h using 100 mM DHS as the substrate (Figure 3A). To optimize the load of the whole-cell biocatalyst on a large scale, a 1 L biocatalysis was performed in a bioreactor using fermentative broth with DHS as the substrate. The DHS conversion rate increased gradually with the increase in whole-cell biocatalyst dose. When 20 $OD_{600}$ of whole-cell biocatalyst was loaded, the DHS conversion rate reached 95% at 1.5 h (Figure 6A). This result indicates that a high concentration of DHS can be rapidly converted into SA by about 20 $OD_{600}$ of whole-cell biocatalyst without supplementing exogenous cofactor $NADP^+$ or NADPH.

GDH oxidizes one molecule of glucose to generate one molecule of NADPH, while AroE uses this NADPH to catalyze the reduction of one molecule of DHS to one molecule of SA. Therefore, the theoretical molar ratio of glucose to DHS or SA for whole-cell biocatalysis is 1:1. However, with the catalytic reaction proceeding, a small amount of glucose may enter the catabolism pathway or a small amount of NADPH generated by GDH may participate

in other intracellular reactions or be decomposed, which leads to an increased demand of glucose. According to previous reports, excess glucose substrate is often added into the reaction system to maximize the catalytic efficiency of the whole-cell biocatalyst [24,26]. To optimize the glucose utilization, we tested the actual demand of glucose during the scale-up whole-cell biocatalytic process. When the molar ratio of glucose to DHS was in the range of 0.8–1.1, the conversion rate of DHS increased gradually with increasing glucose load. However, when the molar ratio was greater than 1.1, there was no significant effect on the conversion rate of DHS. Therefore, the optimum glucose load for efficient whole-cell biocatalysis on a large scale was 1.1 equivalents (Figure 6B). The actual demand of glucose was 10% more than the theoretical demand, suggesting that there was a small amount of glucose loss during the whole-cell biocatalytic process.

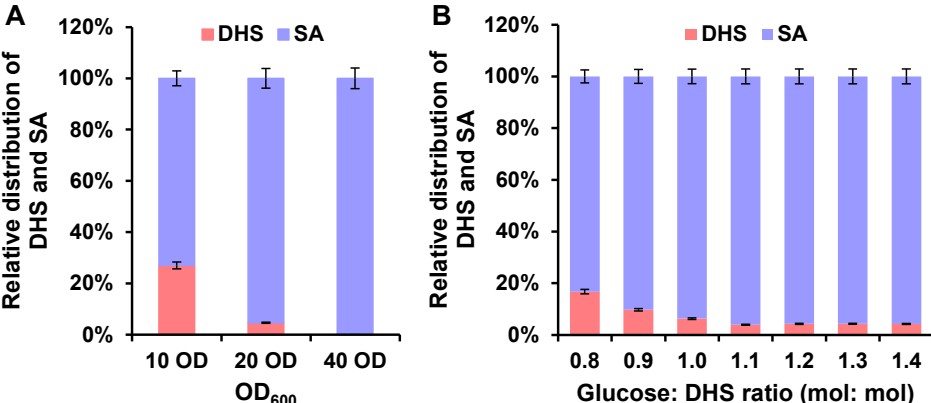

**Figure 6.** Effects of $OD_{600}$ (**A**) and glucose (**B**) on conversion rate of DHS for whole-cell biocatalysis on a large scale. (**A**) The biocatalyst load for efficient whole-cell biocatalysis was tested between 10 and 40 $OD_{600}$ with 1.5 equivalents of glucose; (**B**) the glucose load was examined with various equivalents of glucose (0.8–1.4) using 20 $OD_{600}$ of RSF-serA whole-cell biocatalyst. The reactions were carried out on a 1 L scale at 34 °C, pH 7.0 with 80 g/L DHS, and the concentrations of DHS and SA were determined at 1.5 h. Data are presented as the mean ± standard deviation of two independent experiments.

In order to test whether the whole-cell biocatalyst can be recycled, 20 $OD_{600}$ of RSF-serA whole-cell biocatalyst was employed to verify the hypothesis. Firstly, about 87 g/L DHS in the fermentative broth produced by *E. coli* WJ060 through fed-batch fermentation was completely converted into SA within 3 h (Figure S7A). Then, the RSF-serA biocatalyst was collected by centrifugation and was reused to catalyze the reaction again. The catalytic activity of RSF-serA biocatalyst was significantly reduced, and the SA titer was only 28.1 g/L at 18 h. There was still 21.2 g/L DHS in fermentative broth, and about one-third of DHS was lost during the whole-cell biocatalytic process (Figure S7B). These results indicate that the RSF-serA biocatalyst possesses high catalytic efficiency, but poor catalytic stability, and its catalytic activity can only undergo one round of whole-cell biocatalysis. We speculate that the poor stability of BmGDH and/or AroE or the intracellular $NADP^+$ and NADPH might lead to the poor catalytic stability of the whole-cell biocatalyst.

In order to further simplify the procedure of SA production, DHS fermentative broth containing WJ060 cells was directly used as the substrate for the whole-cell biocatalytic process. About 88 g/L DHS could be converted into ~77 g/L SA within 2 h, which was catalyzed by 20 $OD_{600}$ of RSF-serA whole-cell biocatalyst (~30 g/L wet cells) with 1.1 equivalents of glucose. The average conversion rate of DHS reached 98.4% mol/mol, and the productivity of SA was up to 40.8 g/L/h (Table 1). In addition to the whole-cell biocatalyst and glucose, a large amount of NaOH solution was also required to maintain the system pH at 7.0 because a great deal of gluconate was synthesized by GDH during the NADPH regeneration process. These three components resulted in an increase in broth volume and decrease in SA titer. After removing cell pellets by centrifugation,

the supernatant of fermentative broth contained 81.6 g/L SA (Table 1) with a yield of 23.1% mol/mol glucose. If the fermentative broth with a higher DHS concentration is used as the substrate, the titer of SA would be further improved. Using metabolically engineered *C. glutamicum* as the whole-cell biocatalyst, a titer of 141 g/L SA was achieved from glucose in a 1 L reactor via a growth-arrested cell reaction [1]. However, the whole-cell biocatalyst load was up to 100 g/L and the reaction time was up to 48 h, which was much higher than that of our approach (Table S1). Unfortunately, a large amount of gluconate is produced during the whole-cell biocatalytic process, which is not only not conducive to the purification of SA, but also causes the waste of carbon substrate. Therefore, an efficient NADPH regeneration system without byproduct generation needs to be constructed in a future study, such as efficient NADP$^+$-dependent formate dehydrogenase or phosphite dehydrogenase.

**Table 1.** The titer and yield of SA from DHS by RSF-serA biocatalyst after 2 h of whole-cell biocatalysis.

| Entry | DHS Titer (g/L, with Cells) | DHS Broth Volume (L) | SA Titer (g/L, with Cells) | SA Broth Volume (L) | Yield (mol/mol, %) | SA Titer (g/L, Supernatant) | Productivity (g/L/h) |
|---|---|---|---|---|---|---|---|
| 1 | 89.8 | 2.24 | 78.8 | 2.56 | 99.0 | 83.0 | 41.5 |
| 2 | 87.5 | 2.90 | 76.1 | 3.33 | 98.7 | 80.0 | 40.0 |
| 3 | 88.3 | 3.09 | 76.5 | 3.52 | 97.5 | 81.8 | 40.9 |
| Mean ± SD | 88.6 ± 1.2 | | 77.1 ± 1.5 | | 98.4 ± 0.8 | 81.6 ± 1.5 | 40.8 ± 0.7 |

## 4. Conclusions

SA is the key hydroaromatic intermediate in the biocatalytic conversion of glucose to aromatic bioproducts, as well as an important precursor for synthesizing a variety of valuable antiviral drugs. Intracellular NADPH availability plays an important role in the biosynthesis of SA from glucose via the shikimate pathway. In this study, a series of NADPH self-sufficient whole-cell biocatalysts were designed and tested for the bioconversion of DHS to SA. The enzyme assay and protein expression level analysis indicated that the activity of AroE may be the rate-limiting step for the whole-cell biocatalytic process. An L-serine auxotrophic strain was used as the *Bmgdh–aroE* co-expression host for preparation of the whole-cell biocatalyst. This strategy not only avoids the addition of antibiotics throughout the process of fed-batch fermentation, but also maintains the stable self-replication of expression plasmids in host cells. After optimizing the conditions for the scale-up of whole-cell biocatalysis, the efficient relay-race biosynthesis of SA from glucose by coupling microbial fermentation with the biocatalytic process was finally achieved. A titer of 81.6 g/L SA were obtained from the supernatant of fermentative broth in 98.4% yield (mol/mol) from DHS with a productivity of 40.8 g/L/h (Table 1). This study provides an effective strategy for the biosynthesis of fine chemicals that are difficult to obtain through de novo biosynthesis from renewable feedstocks, as well as for biocatalytic studies that strictly rely on NAD(P)H regeneration.

**Supplementary Materials:** The following supporting information can be downloaded at https://www.mdpi.com/article/10.3390/fermentation8050229/s1: Figure S1. Simple diagrams of the plasmids; Figure S2. Fed-batch fermentation of DHS-overproducing strain WJ060; Figure S3. SDS-PAGE analysis of different GDH proteins; Figure S4. Effects of galactose permease GalP or glucose facilitator Glf on the catalytic efficiencies of whole-cell biocatalysts; Figure S5. The effect of pH on bioconversion of DHS to GA by PET whole-cell biocatalyst; Figure S6. SDS-PAGE analysis of BmGDH and AroE proteins; Figure S7. Bioconversion of DHS to SA by RSF-serA whole-cell biocatalyst; Table S1. Comparison of different catalytic methods for the production of SA; Table S2. The strains and plasmids used in this study; Table S3. Primers used in this study.

**Author Contributions:** Conceptualization, X.W., F.W., and Q.W.; methodology, X.W. and F.W.; validation, X.W., F.W., D.Z., G.S., and W.C.; investigation, X.W., F.W., and D.Z.; data curation, F.W. and Q.W.; writing—original draft preparation, F.W.; writing—review and editing, F.W., C.Z., and Q.W.; visualization, X.W. and F.W.; supervision, Q.W.; project administration, Q.W.; funding acquisition, F.W. and Q.W. All authors read and agreed to the published version of the manuscript.

**Funding:** This research was funded by the National Key Research and Development Program of China (2018YFA0903700 and 2018YFA0901402), the National Natural Science Foundation of China (31700080), and the Scientific Instrument Developing Project of the Chinese Academy of Sciences (YJKYYQ20170023). Qinhong Wang was funded by the Tianjin Industrial Synthetic Biology Innovation Team.

**Institutional Review Board Statement:** Not applicable.

**Informed Consent Statement:** Not applicable.

**Data Availability Statement:** Not applicable.

**Acknowledgments:** Not applicable.

**Conflicts of Interest:** The authors declare no conflict of interest.

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
