# Peer review of "Cofactor Self-Sufficient Whole-Cell Biocatalysts for the Relay-Race Synthesis of Shikimic Acid"

_fermentation, doi:10.3390/fermentation8050229_

Round 1
Reviewer 1 Report
This manuscript written by Wang et al. reported the efficient synthesis of shikimic acid (SA) by the coupling reaction using 3-dehydroshikimic acid (DHS) fermentative cell and AroE-glucose dehydrogenase co-expressing cell as a biocatalyst. They established GDH-expressing SA-synthetic biocatalyst and demonstrated its usefulness as a cofactor regeneration system in the relay-race synthesis with E. coli WJ060, a DHS fermentative cell. The manuscript is well written and provides various useful information for fermentation technology using engineered microorganisms. The reviewer considers that this manuscript would be paid much attention to the reader of the journal Fermentation. However, several points mentioned below should be clarified before publication.
Major point:
The relay-race system needs to transfer metabolites from one to the other cell. In the present study case, DHS is synthesized from E. coli WJ060 strain and then transferred to the aroE-GDH-expressing strain, which converts it to SA. Considering the DHS transfer, how is DHS secreted from WJ060 and is taken up by aroE-GDH-expressing strain? Is there any specific transporter? Did you perform any treatment on the aroE-GDH-expressing cell?
Minor points;
(Section 3.3) The authors checked the expression level of Bmgdh and AroE from pRSFduet-1 using the plasmids that express either of the proteins separately. I do understand it was difficult to distinguish the expression level of both proteins in co-expressed cells by SDS-PAGE because the molecular masses are close to each other, but the expression environment between co-expression and single protein expression in pRSFduet-1 would be significantly different. How do the authors consider about this point?
(Figure5E) In Fig 4B, AroE expression level of Bmgdh-AroE whole catalyst is shown to be eight times higher than BmGDH. However, in Fig 5E, AroE activity in the RSF strain is just about 50 U/mg which corresponds to approximately 2.5 times BmGDH. Why are they different?
(TableS1) About WJ060 strain, did the authors publish about the strain elsewhere? What does the genotype “TTG” mean? The information on this strain seems insufficient.
